# Implementing Rapid Climate Action: Learning from the 'Practical Wisdom' of Local Decision-Makers

**Andy Yuille ***, **David Tyfield and Rebecca Willis**

Lancaster Environment Centre, Lancaster University, Lancaster LA1 4YQ, UK; d.tyfield@lancaster.ac.uk (D.T.); r.willis@lancaster.ac.uk (R.W.)

**\*** Correspondence: a.yuille1@lancaster.ac.uk

**Abstract:** A global goal to limit dangerous climate change has been agreed through the 2015 Paris Accords. The scientific case for action has been accepted by nearly all governments, at national and local or state level. Yet in all legislatures, there is a gap between the stated climate ambitions and the implementation of the measures necessary to achieve them. This paper examines this gap by analysing the experience of the following three UK cities: Belfast, Edinburgh, and Leeds. Researchers worked with city officials and elected representatives, using interviews and deliberative workshops to develop their shared understandings. The study finds that local actors employ different strategies to respond to the stated climate emergency, based on their innate understanding, or 'phronetic knowledge', of what works. It concludes that rapid climate action depends not just on the structures and mechanisms of governance, but at a deeper level, the assumptions, motivations and applied knowledge of decision-makers.

**Keywords:** climate change; politics; local climate action; local government; climate emergency; phronesis; practical wisdom; crisis; UK

## 1. Introduction

A large majority of decision-makers in local, state-level and national governments are committed to acting on climate change. Following the Paris Accords of 2015, the findings of the Intergovernmental Panel on Climate Change in their 'special report' of 2018 [1] and global climate protests and school strikes, many legislatures, including the UK, France, Canada, Ireland and many local governments, made a formal declaration of a 'climate emergency'. These declarations can be seen as an acknowledgement of the gravity and urgency of the climate crisis. They are often accompanied by a 'net zero' target for greenhouse gases (GHGs), reducing emissions as far as possible and compensating for any remaining emissions through the removal of GHGs from the atmosphere.

The stage is set, then, for ambitious climate action. Yet, it is increasingly apparent that a gap is opening up between the stated intentions and the action necessary to achieve them [2,3]. Many explanations have been put forward for this gap, from the view that the fundamental growth imperative of capitalist accumulation is not compatible with the emission reduction [4,5] to more technical analyses that suggest, for example, that the costs of GHG pollution are not currently factored into decision-making, and that economy-wide carbon taxes are required [6]. Yet a recent survey of climate governance research [7] revealed that, while questions of technical governance and policy design have been well studied, there has been less attention paid to the crucial issue of how such solutions might be implemented, and by whom. An earlier study [8] (p. 755) notes that there is a limited understanding of the role and motivations of senior decision-makers (SDMs) in climate governance: "despite the important role of SDMs, many analyses of the climate change problem gloss over them".

This paper thus focuses on a neglected area of research, which is the lived experience and understandings of local decision-makers. Building on earlier work examining the

attitudes and motivations of national politicians [9], the paper reports on a study of officials and elected representatives in three UK cities, Belfast, Edinburgh and Leeds. Researchers worked with local decision-makers to learn from their practical skills and know-how, as individuals responsible for the implementation of ambitious declarations and targets.

The paper begins by discussing the theoretical approach that informs the methodology and analytical orientation of the research, emphasizing the importance of 'phronetic knowledge' [10], or implicit understandings that professionals have about the possibilities and constraints of their role. These insights are discussed in detail in Section 2 below. The theoretical orientation of this study informs the method, which is described next: qualitative, semi-structured interviews with officials and elected representatives in each city, followed by small deliberative workshops offering researchers and city stakeholders the opportunity to reflect together on the study's findings. This allowed the academic researchers and policy professionals to learn together, with both providing their own contextualised expertise. Thus, all involved could reflect, collaborate and innovate their practice. Data from interviews and workshops were analysed iteratively, through two levels of analysis. First, the stated barriers and enablers of climate action were drawn out. The second stage was an inductive identification of distinct patterns in participants' working strategies and their interactions with each other.

Turning to the results, at the first level of analysis, the study confirmed that climate change now has a high political priority across the three cities, and that this has significantly increased in recent years; however, a gap is opening up between the stated ambitions and the plans to achieve them. A particular problem is the lack of clarity over the division of responsibilities between the national and local governments, and, given the UK governance context, the lack of powers and resources at a local level, which limits the ability of local areas to act. Whilst participants tend to stress the importance of collaboration between officers and politicians, between different departments and between political parties, the research finds that perspectives on the same issues varied, often radically, within an organization.

At the second, deeper level of analysis, the focus shifts from the *what* of the challenges, opportunities, progress and barriers relating to rapid climate action, to *how* these factors are engaged with as part of the participants' everyday working lives. The patterns in working practices were identified and categorised, as follows:

- 'Crusaders' seek to embed climate action in the council and beyond, seeing their role as 'getting the message out' and 'changing the culture'. They work strategically to establish climate action as an urgent priority that cannot be sidelined, and to shift the accepted range of what is achievable;
- 'Entrepreneurs' use their knowledge of the existing concerns, situations and ways of working to seek out opportunities to promote climate action. They look for synergies with existing programmes and priorities and try to link the strategic to everyday routines and decisions;
- 'Pragmatists' recognise the importance of the climate agenda, while retaining a strong focus on pre-existing personal and council objectives and how strategic ambitions can or cannot be delivered through existing procedures;
- 'Weavers' focus on collaboration and connections, aiming to mesh together agreed-upon high-level aims with the specific contested measures needed to achieve them, and to build and maintain trust and support.

The study concludes that the successful development and implementation of climate strategies at local level will require a sustained focus, not just on what needs to be done, but on how local decision-makers can find their way through a conflicting and sometimes contradictory governance landscape.

## 2. Methodology

### 2.1. Theoretical Orientation

The research presented here is informed by a particular theoretical orientation that emphasises the importance of understanding the context of decision-making within institutions, and particularly the lived experience of the individuals involved. This is explained below, with reference to the following three linked concepts: transductive analysis; 'phronetic' knowledge; and understandings of crisis, particularly relevant to the government of 'climate emergency'.

As outlined in the introduction, many national and local governments have committed to ambitious climate strategies, but a gap has opened up between those ambitions and their implementation. In analysing this gap, scholars tend to adopt one of two broad methodologies, which may be loosely characterised as 'realist' or 'idealist'. In both cases, research proceeds by exploring what is currently the case and why, before suggesting solutions to the problems thus defined. In realist analysis the entire enterprise, and especially the suggestion of ways forward, prioritises what seems 'realistic' given the understanding and problem definition. This thus includes the vast majority of (social) scientific literature on climate action. Idealist analysis, conversely, tends to adopt a more explicitly normative stance in both its problem diagnosis and subsequent recommendations. Its analysis may thus tend towards 'deeper' causes and be less focused on specific issues, measures or interventions. For instance, it may offer a system-level critique of current political and economic practices, asserting that only change at this system-level can achieve climate ambitions, and advocating a new political or social settlement [4,11]. Despite their very different orientations, though, the realist and idealist approaches have a shared tendency for their desire for solutions to dominate the whole exploration, setting up a scramble to reach these conclusions. These ways forward also emerge as a final speculative jump in the analysis, with the proceeding analysis the 'run up' supposedly on sure ground.

We argue, however, that the nature of the climate crisis requires a much fuller and enduring examination of the state we are in. The climate emergency is a profound crisis, challenging settled common senses and ways of thinking and doing [12,13]; an experience perfectly exemplified by the dissonance arising from acknowledging, or even actively campaigning for the declaration of climate emergency—and then not knowing what to do about it. Such sentiments are evident, and increasingly documented [14] across all tiers and forms of government, as well as in business and civil society.

Such paralysis urgently invites a different approach to the problem. The fact that so many interested and committed decision-makers find themselves in this situation suggests prevailing modes of seeking ways forward offer little guidance. Rather than directly searching for solutions, as realist or idealist positions might, this paralysis invites a shift towards examining the predicament itself with a strategic lens, going *into* it to illuminate what is already the case. This can be called a 'transductive' form of analysis, drawing out lessons by going *through* a careful and strategic exploration of the present. It can be contrasted with the more familiar deductive or inductive forms of reasoning—or, indeed, abductive or retroductive forms [15]—that typify realist or idealist approaches, which draw on evidence to formulate lessons in the light of what is already deemed known and/or wanted.

Dominant approaches to analysis necessarily ground such learning foundationally in existing cognitive frameworks. Yet confronted with paradox and paralysis, it is precisely these latter that are the problem. In contrast, transductive analysis opens up rethinking of existing paradigms by diving deeper into an appreciative [16] and pragmatic characterisation of the predicament, to explore and reflect upon how things are currently constituted, how they 'work' or 'fail' (and for whom), and how different actors, and their practices, are implicated.

In this approach, a key to effective mobilization of settled paradigms is an emphasis on learning from the (possibly tacit, likely contradictory) understandings of those embedded within the situation under study, focusing on practical wisdom or 'phronesis'. The

term 'phronesis' refers to an Aristotelian categorisation of knowledge, described as "the practical wisdom that emerged from having an intimate familiarity of what would work in particular settings and circumstances" [17] (p. 369). As such, it can be distinguished from episteme (universal knowledge) or techné, (practical application of knowledge). In a recent revitalization of this concept, though, this practical wisdom also has specific concern for the irreducible power relations that condition, and are constituted by, any body of knowledge [18].

Following this approach, this study aims to uncover the phronetic knowledge of those active in local government as a promising avenue to break the current deadlock in climate action at this key scale. It scrutinises their innate understandings of the possibilities and constraints of their role, or what Sanford Schram and colleagues refer to as "the 'unconsciously competent' expertise that ought to be part of the scholarly endeavour" [17] (p. 371) (see also [19]). Thus, the study extends beyond the *'what'* of the challenges, opportunities, progress and barriers relating to rapid climate action, to focus primarily on *how* these factors were engaged with as part of participants' everyday working lives, drawing on their lived experience of addressing them within particular cultural, historical and institutional contexts. This then encourages imaginative, engaged and constructive reassessment by actors themselves of how they are currently working and what they could do differently for more effective pursuit of their goals. Focusing on what is already the case and the opportunities to draw on a 'bricolage' of what is already 'to hand' in novel combinations [20,21] also then enables optimal practicality in the ensuing insights, not least by 'working with the grain' of existing power relations.

It is particularly important to use this phronetic approach in this case because of the profound nature of the climate crisis, or climate emergency. For this is not just a matter of environmental limits, but also links to crises of an emergent society [22] that is new in multiple and profound ways, e.g., as planetary [23], global and cosmopolitised [24,25], and constituted by post-human technologies and nature-cultures [26,27]. Our ability to move forward on climate change is intricately linked to our confidence in the ability of the government—both national and local—to understand its predicament and use its powers wisely. Yet in such unprecedented circumstances this capacity is significantly attenuated.

Following Debray [28], political economist Bob Jessop argues that crises are "complex, objectively overdetermined moments of subjective indeterminacy, where decisive action can make a major difference to the future" [29] (p. 247). They are 'objectively overdetermined' in that there is no single definitive set of causal factors that can be identified as the 'root cause' of the crisis. They are 'subjectively indeterminate' in that there are multiple interested interpretations of the crisis, many of which may yet be proven 'correct' depending on the course of action taken in response and which groups are thereby empowered. In both respects, then, there is no single objective fact of the matter for a standard realist/idealist approach to illuminate first, before handing it over to practical decision-makers. The irony is thus that in such a moment of crisis, seeking the refuge (perhaps with urgently renewed zeal) of the sure grounds of conclusive fact tends only to exacerbate epistemic disorientation and so practical paralysis.

In crises, therefore, a different and strategic approach, working from existing tacit practical understanding, is not only more promising but arguably essential. The importance of this approach regarding the governance of climate emergency is even more striking once we admit, as above, that climate crisis overlaps and threatens to feed into other forms of system crisis, including crises of authoritative knowledge, of state institutions and even of governmental legitimacy. In short, the paralysis on the governance of climate emergency is tenable only for so long, and with the stakes high indeed, not least for the very institutional preconditions of any hope of effective climate action.

In this research, therefore, we work with those actors who are in the midst of responding to the 'climate emergency' and specifically burdened with governmental responsibility to do so. We make explicit their phronetic understanding both of the crisis and of the

institutions that are required to respond, in order to help with finding locally appropriate ways forward.

*2.2. Data Collection and Analysis*

In keeping with the theoretical orientation described above, the study methodology was designed to elicit the 'phronetic knowledge' of officials and politicians in local government, across the following three UK cities under study: Belfast, Edinburgh and Leeds. These cities form part of the Place-Based Climate Action Network (PCAN), a network of academic, policy and business stakeholders, co-ordinated by the London School of Economics and dedicated to translating climate policy into action 'on the ground'. PCAN provided the funding for this project. All three cities are covered by the UK Government's legally binding national target of achieving net zero by 2050. This target was introduced in June 2019 in an advance from the previous target of at least an 80% reduction from 1990 levels by 2050. The UK Government does not at the time of writing have a specific climate strategy, rather its approach to emissions reduction is embedded across a range of different strategies, e.g., industrial strategy, clean growth strategy, resources and waste strategy, and more recently a 'ten point plan for a green industrial revolution'. As cities in the devolved nations of Northern Ireland and Scotland, respectively, Belfast and Edinburgh also have a tier of devolved national government between the city council and the UK Government, with their own climate-relevant policies. The Scottish Government has set a net zero target for 2045, while Northern Ireland does not have a nation-specific emissions reduction target, but must contribute to achieving the overall UK target. Leeds (in England, which does not have a devolved national government), is only covered by UK Government policies.

Local governments have a range of statutory responsibilities with significant impacts on emissions (e.g., spatial planning, transport planning (except in Northern Ireland), waste collection and disposal, education and social housing provision (both involving an estate of largely older, energy inefficient buildings) and social care (involving substantial travel by employees). They also have a range of non-statutory responsibilities, e.g., provision of leisure and recreation facilities, including parks and green spaces. As well as cutting emissions from their own estates and operations, some councils (including the three cities in this study) are also putting in place strategies, mechanisms and targets for reducing emissions from the territory that they govern as a whole, including the public and private sector and household emissions. One mechanism to facilitate this are PCAN 'Climate Commissions', which bring together key city stakeholders with the council to collaborate on city-wide emissions reduction.

While climate change has featured as a consideration in some local government activities for several years (e.g., transport and spatial planning), it is only very recently that this has become a significant priority for them, and in most cases local climate plans and strategies are only now emerging, so their impacts cannot yet be quantitatively assessed. The climate strategies of each city council in this study have been informed by 'mini-Stern reviews', which assessed the cost and carbon effectiveness of a wide range of the low carbon options that could be applied at the local level in households, industry, commerce and transport, and explored the scope for their deployment, the associated investment needs, financial returns and carbon savings, and the implications for the economy and employment. However, even in those areas where the local government has responsibility (e.g., transport and spatial planning), they operate within a policy framework set by the national and devolved governments, which constrains their scope for action.

In Table 1 we set out some of the relevant organizational and governance context of the cities in this study.

The phronetic approach is not intended to assess the appropriateness of these strategies or targets, nor to measure progress towards implementing or achieving them, but rather to explore how, in practice, key actors within these institutions engage with them and integrate them within everyday working practices that are also subject to a wide range of other pressures, priorities and responsibilities.

**Table 1.** City councils' organisational and governance context.

| | Belfast | Edinburgh | Leeds |
|---|---|---|---|
| Climate emergency | Declared October 2019 | Declared May 2019 | Declared March 2019 |
| City-wide emissions target | Quantitative target due to be set in 2021 | Net zero by 2030 | Net zero by 2030, with emissions halved by 2025 |
| Strategy in place | City-wide net zero roadmap published December 2020 (aligned to UK 2050 target) | Immediate action plan to reduce council emissions published October 2019; city-wide net zero roadmap published December 2020 | City-wide net zero roadmap published April 2019; updated January 2020 |
| Political context | Neither nationalist (supporting a united Ireland), nor unionist (supporting continued British rule of Northern Ireland) parties have a majority. | Council controlled by a minority Scottish National Party/Labour coalition, with the Conservatives the largest party in opposition. | Council controlled by one party (Labour), with a large majority. |
| Climate Commission | Established January 2020 | Established February 2020 | Established September 2017 |

The methodology followed an iterative process, beginning with interviews, which were coded and analysed, with findings presented back to small groups of stakeholders through online workshops, where they were discussed and refined. The workshops were then analysed, and conclusions and recommendations drawn out. Each of these stages is described below. All interviews and workshops were carried out online. Ethics approval was sought and granted by Lancaster University Faculty of Science and Technology Ethics Committee. The interviews and workshops were held under conditions of anonymity, so the data from this study is not available as a dataset. Given the small sample size and the need for anonymity, we have not attributed the quotes used in the results section in any way.

Interviews: Five individuals from each city were recruited, following advice from local researchers involved in PCAN. In each city, we interviewed two elected representatives, i.e., local politicians, and three senior officials. In each city, we included those who had direct responsibility for climate strategy (for example, a politician chairing a climate working group or an official responsibible for council-wide sustainability policy) and those working in areas where policies are necessary to drive carbon reduction, such as transport, planning, housing and economic development. Interviews were qualitative and semi-structured, using a narrative approach [30], and were conducted as an exploratory conversation. Participants were offered anonymity. The interview questions covered the responsibilities of that individual's role, their sense of 'what works' in policymaking, their views on how climate change is factored into policy decisions, and what is needed to allow rapid climate action. Interviews were recorded, transcribed and coded using the Atlas-ti programme.

Workshops: Findings from the interviews were summarised into a discussion paper. Through the PCAN network, each city was offered the opportunity of a workshop. Workshops were held in two of the three cities, Belfast and Leeds, with Edinburgh stakeholders declining the offer due to resource issues during the COVID-19 pandemic. At the workshops, the findings of the interviews were presented, and small-group discussions and creative visualisation techniques were used to encourage reflection and phronetic learning on the part of all participants, including the researchers. Participants compared the differing viewpoints held by different actors, in order to develop a deeper understanding of how climate-relevant decisions are framed and made, and to put forward proposals for change. Participants included interviewees, other city stakeholders, PCAN team members, and the project team, with 12–15 participants in total in each workshop. The workshops were held online, using the Zoom platform.

Engagement and dissemination: An anonymised report of findings was prepared, looking at the specific circumstances of each city, and drawing out commonalities. This was disseminated through a webinar, through the PCAN network and through networks in each city. Results are summarised in the following section.

## 3. Results

The theoretical orientation of this paper informs the presentation of the results. We start by reporting the 'what' of responding to the climate emergency, which includes the enabling and constraining factors reported by the participants. We then dig deeper into the 'how', exploring how the participants understood and navigated these factors, and what strategies they employed to try to make progress against their own goals and those of their organisation, in the context of responding to the climate emergency.

### 3.1. Enabling and Constraining Factors—The 'What'

Across the three cities, common themes emerged from the interviews with both officers and politicians to explain the progress they had made, the scale of the challenge still facing them, and the difficulties and opportunities they encountered or foresaw in pursuing their locally agreed climate strategies and targets. These themes were confirmed and developed further in the deliberative workshops. They are summarised at a very high level in Table 2 and developed in the following text, illustrated with direct quotations from participants (inset and italicised).

**Table 2.** Key enabling and constraining factors.

| Theme | Key Finding |
|---|---|
| Political priorities | Climate now higher up political agenda |
| Ambition v implementation | Limited understanding of how climate goals will be achieved |
| National-local interaction | National policies constrain local action |
| Organisational culture | Alignment on climate agenda not yet established within or beyond city councils |
| Framing the issue | Need to present climate action as mainstream choice with multiple benefits |
| Devil in the detail | High-level support undermined by contentious local implementation |
| COVID-19 risks/opportunities | Opportunity to 'build back better', but risk of return to 'economy first, environment later' model |
| Place-based approaches | Climate action understood as locally specific and embedded in meaningful and symbolic elements of place |

Political priorities: Climate protests, widespread media coverage, and increases in vocal public concern have shifted what is possible and necessary for councils to do, triggering the declaration of climate emergencies and creating a context in which the development of local emissions reduction targets and strategies to achieve them became widely accepted as important. Public support and strong senior political and officer leadership on climate action are vitally important, especially when particular policies or initiatives might be unpopular.

> *"The profile given to climate change has removed the scales from some people's eyes or elevated it in terms of their political priorities."*

Ambition vs. implementation: Despite the ambitious commitments set out in the locally agreed strategies, and a broad understanding that every aspect of the council's work would have to align with the net zero target, there was no real understanding of how this agenda would be incorporated into service delivery plans and reported against. There was a shared feeling of moving into uncharted, experimental waters [31].

*"The price for most officers [of an ambitious target] is we can't see the path to that."*

National–local interaction: The councils have huge potential to drive emissions reductions but currently lack the powers, funding, and statutory responsibility to do so. Policies and procedures, often nationally imposed (e.g., in planning, housing and transport) severely restrict the local government's ability to prioritise carbon reduction, linked to a perceived lack of national leadership [32].

*"The main dilemma for any local authority is, none of this is statutory. We have no piece of legislation that says we need to do this."*

Organizational culture: A highly collaborative and aligned approach prioritising the decarbonisation agenda is needed both within the councils (between officers and politicians, different departments, and political parties) and with wider stakeholders [33,34]. This approach was not yet in place, although the PCAN Climate Commissions were perceived as making positive progress. However, radically divergent perspectives remained even within institutions, e.g., on whether a council's strategic framework helped or hindered delivering the climate agenda, or whether local political configurations have made progressive action easier or harder. External expertise and partnerships were often seen as key drivers in pushing councils to develop ambitious decarbonisation commitments [35].

*"Even though there is a lot more consensus now than there was even two or three years ago, we're still not necessarily all pointing in the same direction."*

Framing the issue: Framing climate action [36] in terms of its co-benefits—e.g., reducing fuel poverty, generating jobs, and improving air quality and public health—is an important route to securing political and public support [37,38]. To achieve significant change, climate action has to be understood as the best, mainstream course of action/investment, rather than a 'green alternative'. Politicians and officers agreed that once the high-level political decisions have been made, officers need to present and frame evidence and options that will help politicians make the 'right' choices in that context; both officers and politicians have political agency [39].

*"... positioning green action as just the best action to take. Not green, but actually the best choice... So the risk of not doing this is greater than the cost of doing it; the opportunity of it is greater than the uncertainty you face right now."*

Devil in the detail: The widespread political, officer and public support for the 'big ideas' of tackling the climate emergency can quickly be reversed in specific, contentious instances, e.g., public resistance to reallocating road space, or political decisions that support jobs but increase emissions. This can derail individual projects, and cumulatively threaten the achievement of targets, take up significant officer and politician energy, and generate aversion to future interventions.

*"It's the difference between, a lot of people take on board the overall concept that we need to do something about it but they're not necessarily taking that ownership or making that change themselves. I think that's where we struggle to get buy in and support."*

COVID-19 risks/opportunities: The recovery from Covid-19 provides opportunities to drive change, building on learning from the response to the crisis and the potential of the stimulus package, but it also risks the re-emergence of an 'economy first' approach. As emissions reduction is not a statutory duty for local government, it has been squeezed as an objective by austerity [40] and is likely to be even more so during the recovery from COVID-19.

*"As we're beginning to think about coming out of COVID and recovery, people are saying the right things: we don't want to go back, we want to build back better, this is an opportunity ... [but] saying it and meaning it when jobs and growth are in question are two different things."*

Place-based approaches and narratives: A theme that emerged specifically from the workshops was an emphasis on climate action as locally specific. Appeals to abstract or

generalised quantifications, and high-level science and policy, were not seen as persuasive to publics or policymakers. Driving rapid change through locally determined strategies requires making explicit connections with meaningful and symbolic elements of place and developing culturally specific narratives, as well as reflecting spatially specific challenges and opportunities; the city's relation to climate action must be understood and communicated in terms of a lived place as well as an abstract space [41]. This required local government leadership across a range of place-based dimensions, and a concomitant need for associated powers and resources to be devolved at the local level. This is not a matter of 'glorifying the local', but emphasises that implementing rapid climate action in specific locations is not simply a matter of applying national targets at a local level.

> *"We are going to have to tell compelling attractive locally understandable stories about climate action, and we can't just depend upon the language of science and science-driven targets and policy deadlines, these will not land . . . [we need a] way of indigenising this, localising it and using colloquial language and stories . . . to make this really local and tangible for people. I do think that the policy and science stuff, we need it but it ain't going to sell it."*

### 3.2. Personas and Strategies–The 'How'

The results described above broadly aligned with the authors' expectations from contextual background research, experience of working with local government in various professional capacities, and existing literature [42,43]. However, as described in Section 3 above, the phronetic approach taken in this study calls for the analysis to extend beyond the *'what'* of the challenges, opportunities, progress and barriers relating to rapid climate action, to *how* these factors were engaged with as part of participants' everyday working lives. Insights about the 'how' can then in turn re-situate the insights about the 'what' and make them available to be engaged with differently.

As set out above, iterative analysis of interview transcripts led to the inductive identification of distinctive patterns in the participants' working practices, and their interactions with others in relation to local strategies and targets for climate action. We categorised these patterns into four 'personas', as set out in Table 3. These describe the ways in which people engaged with the problem of rapid climate action, which we labelled as crusaders, entrepreneurs, pragmatists and weavers. We followed the principles of phronetic social science in developing these by focusing on the context of specific situations, the values held by the actors embedded in those situations, the production and distribution of power (or agency) in those situations, and generating an active dialogue with relevant publics (interviewees and other key stakeholders within the councils and cities) [44].

**Table 3.** Ways of engaging with the climate agenda.

| Persona | Defining Characteristic |
| --- | --- |
| Crusader | Seeks to establish climate action as an urgent priority |
| Entrepreneur | Seeks to integrate climate with existing programmes and priorities |
| Pragmatist | Seeks to deliver climate action within existing policy and procedural framework |
| Weaver | Seeks to build widespread support for climate action |

These personas were not articulated explicitly by interviewees themselves, but were drawn directly from their accounts of lived experience and their practical responses to concrete situations 'on the ground'. The presence of these categories as distinct and recognisable ways of working, and the practical utility of using them as a framework to think through individual and institutional responses to the challenge of climate change, were confirmed by participants in the deliberative workshops, as follows:

> *"The typology is really helpful because . . . as we think over the next few years about this, we need to be really focused on do we have enough of that mix."*

*"Those characterisations did really resonate with me when I think about types of people we work with in the council and how things are now . . . it then presents the opportunity to be able to understand why someone's behaving like that, potentially moving them into different ways of thinking and . . . bake things in more effectively."*

The personas were enacted in the responses of both the officers and politicians. Individuals may enact different personas at different times and in different circumstances, although they may have a disposition towards performing one or more particular personas. They may be consciously decided upon as a strategy, observed by individuals themselves but without prior intent to act in that way, or performed unreflexively. Below we provide a brief summary of the main characteristics of each persona, exemplified by illustrative quotations from the participants.

Crusaders see their mission as embedding rapid climate action in the work of the council and beyond. They work at a strategic level, within and/or across departments and portfolios as well as with external stakeholders. They see their role as 'getting the message out' and 'changing the culture', driving a shift in strategic focus in order to establish climate action as a real and urgent priority for action that can't be ignored, sidelined or compromised away. They attempt to shift the so-called 'Overton window' of policies and actions that are politically acceptable [45]—although their main target audiences are policymakers and other influential stakeholders, rather than the general public.

*"I'm plugging away at that and that's going to take me a while to get that change to really be embedded in but it's a drip drip. I've got to persuade the officers in the council, I've got to persuade the elected members, I've got to persuade other people."*

However, 'crusading' language and action can also alienate audiences and risks the crusader being seen as disconnected from the mainstream, which can reduce their scope for impacting policy agendas—and fear of this can constrain people from adopting 'crusading' stances [9]:

*"The approach is often counterproductive as well, I sometimes feel. The kind of campaigning, crusading approach sometimes can end up either boring people or alienating people, I guess."*

Entrepreneurs are agile and use their knowledge of existing ways of working, agendas and situations to seek out opportunities to promote climate action. They look for synergies with existing programmes and priorities, and show how they can be delivered together with climate action. They try to link the strategic to everyday routines and decisions, and try to address or avoid the obstacles of implementation in sometimes indirect ways. They tend to operate within the existing 'Overton window' to find openings that can be used to further the climate agenda.

*"How we weave the climate into that, in terms of that being perceived as an opportunity and a positive thing."*

Such an approach, however, runs the risk of climate action getting 'lost' and diluted in amongst other priorities. It can generate a sense of climate being just another factor to be added to existing activity, rather than an existential threat:

*"If you politically mainstream it that de-radicalises it, which is good because it means more people get around the table. But my sense is that within the policy articulation of this it's seen as, 'Oh, it's a normal policy process', when it is anything but."*

Pragmatists recognise the importance of climate action, but also maintain a strong focus on pre-existing objectives and may resist what they perceive as the colonisation by climate of other agendas. They are often engaged with the details and decisions around the implementation or scrutiny of policy impacts, and have a strong focus on process and procedure.

*"My team do get quite frustrated that what seems like a good idea and gets put into a strategy isn't really thought through with all of the consequences because they're not*

*responsible for that delivery side. It's easier to write a strategy that sounds good without actually then having to think about how it gets implemented."*

This persona was particularly identified as potentially generating barriers to action through a reliance on policy frameworks, procedures, and established custom and practice, which may take a long time to change in line with institutional ambitions.

*"You've got senior civil servants who are dead competent civil servants but they're to a person they're pragmatists. So unless there's something that makes them change what they want to do or what they have to do, they're not going to change."*

Weavers focus on collaboration and connections between levels (macro and micro) and between stakeholders (within and external to the council). They aim to mesh together easily agreed high-level aims with the disputed and contested concrete measures needed to achieve them. They are concerned with building and maintaining trust and support (from publics, politicians, officers and other stakeholders). They bring together ideas, approaches and people that may otherwise conflict and attempt to 'weave' solutions from the threads of otherwise potentially disparate positions. As such, they strongly resemble the 'bricoleurs' theorised by Cleaver [21], as key agents in the bricolage of disparate but extant factors into practicable and constructive ways of working.

*"You draw those other stakeholders in, in multiple different ways into the conversation . . . so that policy is something everyone feels they collectively own."*

However, this persona also has the potential to slow action down, as gaining and maintaining broad-based support is inherently time-consuming, and may even serve to underline the tensions between essentially agonistic positions [46].

*"We can get bogged down in years of community consultation and dealing with objections. Each issue gets magnified and sucks more and more energy and time into that, rather than just doing it."*

## 4. Discussion

### 4.1. Using This Typology to Bring about Change

At an institutional level, we suggest that the performance of each of these personas is necessary within local government to drive rapid climate action, despite their potential to generate or amplify the barriers to action. By bringing these personas to presence, and paying attention to the different roles and functions they perform, we make them available to individuals and institutions to adopt, adapt and combine as conscious strategies to help achieve rapid climate action in the context of locally agreed plans and targets. They can be intentionally combined at institutional and individual levels in situation-specific ways, and used as a lens through which to understand and respond to the actions of colleagues and other stakeholders, and to relations within and external to the council.

*"You need to take all of those approaches depending on who your audience is and the tailoring process that you need to adopt to really speak to them and to get across what it is that you need to do. I think it's incredibly useful to set out those different areas, those different approaches."*

*"It certainly would help me to think, as a senior leader, about how I can influence people's thinking and behaviour and potentially use this as a way to help them understand how they're working and encourage them to think in different ways."*

This typology provides a conceptual framework to reflect on and deliberate both organisational strategy and personal effectiveness, responding to the imperative that "social science is arguably practiced best when it produces knowledge that the people being studied can themselves use to address better the problems they are experiencing" [47] (p. 15).

It also highlights the crucial, but often neglected and/or undervalued, role of the Weaver and its importance in generating ways forward, perhaps especially when 'nor-

mal' established processes are proving inadequate. All three of the other roles are readily conceptualised and so adopted by individuals in their working lives, likely with institutional approval. In goal-oriented organisations, and for goal-oriented, committed staff, however, the Weaver may often appear ineffectual in both the vision ('big picture') and the implementation ('brass tacks'). Yet, precisely as bricoleurs [21], they are in fact crucial in navigating institutions through moments of systemic crisis, when new thinking is needed and yet things must also carry on uninterrupted.

Using these personas in these ways—as a lens for understanding current thinking and behaviour, and as part of a deliberate strategy to achieve change—may assist individuals and institutions within local government to overcome some of the more tacit, oblique resistance to rapid climate action that persists internally and externally. None of the participants expressed opposition to acting on the climate agenda, and many noted that overt opposition was now rare, which in itself marks a significant and recent change.

> *"I think in terms of the wider public discourse we see less and less of people vocally being politically against action on these issues."*

Nevertheless, more tacit resistance, or brakes on change were described as widespread within the councils:

> *"I don't think it's all there, political buy-in . . . I can assure you a lot of the council officers I deal with on a daily basis have not bought into it."*

We have categorised these 'brakes', which appear to be motivated by the desire to protect other matters (rather than a resistance to climate action per se) into the following three different modes: direct, indirect, and attributed.

Direct brakes were understood as articulations that prioritising rapid climate action will necessarily be detrimental to other priorities, or that targets are unachievable.

> *"We will be carbon neutral by 2030. Well that ain't going to happen. We could be carbon neutral by 2030 but we'd also be bankrupt. But we might get 85% of the way, sensibly. So maybe 2040 or 2043 might be a more sensible guideline."*

Indirect brakes were understood as articulations that rapid climate action is not possible within established policy and procedures, that strategic ambition has not permeated down to operational processes, or that ambition does not take adequate account of the practicalities of implementation.

> *"The strategy says we're going to have a million trees or something like that, what does that actually mean? . . . there's no additional resource for any of that but there's just an expectation that we'll pick it all up."*

Attributed brakes were understood as articulations that change will be impeded by the actions or attitudes of others (publics, politicians, officers or businesses), which in itself generates resignation that change will be delayed or diminished.

> *"In spite of the great words of the vision, there's practical things on the doorstep . . . They see the big stuff but they act on the small stuff and the small stuff they act on is often contradictory to the big stuff."*

The cumulative effect of these brakes on change was summed up by one participant as follows:

> *"It's that non-decision making, or the quiet opposition, or the lack of active support, which I think is probably the undercurrent which is really stopping some of this from moving forward as quickly as it could."*

We thus offer these interpretations as a contribution to a phronetic approach to social science, both in that it draws on the participants' own phronetic understandings or practical wisdom about delivering rapid climate action, and in that it is intended as an intervention, the production of contextually specific knowledge that aims to enhance that phronetic understanding and help people act more effectively in particular situations. In both respects,

we highlight that the importance and existing practice of the four personas, including the crucial bricoleur role of the weaver, emerged immanently from our interviewees' own experience.

Using this typology of personas as a lens to think through how to address these brakes on change, and how to engage with the factors highlighted in Section 3.1, can provide a new perspective on the methods and tactics needed to deliver rapid climate action. For example, at an institutional level it could be used to map and develop competencies, aims and ways of working within and across departments, and to shape corporate and communication strategies on climate action; at a team or individual level it could be used to scope out the most appropriate ways of working to engage at different stages of a project and with different stakeholders, to identify gaps in capacity or adjust operational tactics.

One way of operationalizing these personas in considering either an individual or organizational strategy could be to characterise them in terms of their primary focus (i.e., whether they focus on what is to be done, or on how it is to be done) and their primary concern (i.e., developing policy/goals, or considering the implications of policy implementation). This classification can be presented as a matrix (Figure 1), which can be used to think through which arrangements and combinations of characteristics might be needed, or lacking, in a given situation.

| | | Primary concern | |
|---|---|---|---|
| | | *Goal/Direction* | *Implementation* |
| **Primary focus** | *What* | Crusader | Pragmatist |
| | *How* | Entrepreneur | Weaver |

**Figure 1.** Plotting primary concern against primary focus of typology personas.

This approach may be useful in helping to think through how to bridge the gaps between ambition and implementation, and between immediate actions and long term goals.

*4.2. Bridging the Gaps: The Difficult Middle Ground*

An extended quote from one participant on the process of making a climate emergency declaration and setting a net zero target helps to understand the gap between these ambitious, high-level statements and the everyday reality of local politicians and officials:

> *"That all happened in a number of weeks, going from right, we want to be really meaningful and radical in this and we've got political sign-up to work out what that looks like, to the external environment is requiring us to jump straight to a target that we have no idea how to get to, no evidence as to whether it's the right thing whatsoever, apart from a load of experts telling us that's what needs to happen if we're to take the climate emergency seriously. So while we've been on that path to get there, we probably wouldn't have got to 2030, we were pitching 2037 as radical, the politics overtook us and gave us that target."*

Taking the climate emergency seriously thus requires fundamental and previously unimaginable change, affecting every decision, across all service areas, internal operations and external procurement, as well as extensive stakeholder engagement to bring down local emissions outside of the council's control. This is a daunting prospect, which unsettles deeply sedimented and institutionalised common senses that specifically serve and enact existing concentrations of power. Despite the superficial recognition of the magnitude of change required, and considerable personal, professional and political commitment, its practical implications are not yet grasped and, perhaps more significantly, the path to understanding and engaging with these implications is very far from clear.

Several workshop participants likened the climate agenda to the equalities agenda, to emphasise an analogous need to mainstream and embed the climate agenda, or to propose an analogous method for doing so (e.g., to require climate impact assessments for all decisions). However, this analogy could also be taken as an illustration of the scale and nature of the challenge. Despite anti-discrimination laws dating back to the 1960s, and policies and mandatory procedures in national and local governments, such as equality impact assessments, very few people would argue that the equalities agenda has been satisfactorily addressed and that discrimination is no longer a problem. (We also note, with thanks to an anonymous referee, that such considerations could also be said to apply to the wide range of intersecting issues associated with, say, delivery of the sustainable development goals.)

Likewise, achieving the ambitious emissions reductions, to which local governments are committing themselves and their cities, will require more than plans, policies, technologies and targets. They will require an unprecedented cultural shift such that it is no longer acceptable (much less common sense) to view, for example, jobs or economic growth as a trade-off against emissions reductions. Rather, emissions reductions will have to become the lens through which other priorities can be achieved, they must be pursued together rather than set against each other. Generating such a shift, we suggest, requires the kind of phronetic approach that we have gestured to in this paper, and which this paper makes a small contribution to by exploring not just *what* challenges local politicians and officials face, but also *how* they engage with them in specific lived contexts.

This approach thus calls for both researchers and practitioners to 'dive into' the predicament and attempt to understand it from the inside, rather than simply seeking solutions that can be externally imposed upon it, drawing on both the explanatory, 'objective', realist registers associated with detached problem-solving, and the often-ignored performative, narrative, constructivist registers associated with a more interpretive understanding. The challenge is fundamentally one of ongoing, practical learning, not just about the objectively overdetermined external conditions of the climate crisis, but also about the subjectively indeterminate responses to crisis; this includes the interpretations of the crisis and how these are (or are not) integrated into the rich complexity of everyday lives. This involves the practice and incubation of a situated strategic–ethical wisdom— phronesis—at individual and thence collective and societal levels, in which this learning practice and orientation is both the means and end, practice of and capacity for, sustainable transition [48].

This kind of approach, guided by pragmatic choices and embedded in an ongoing iterative process of learning and doing, is necessary to bridge the gaps between ambition and implementation, and between the immediate and the long term. The participants in this study felt clear about the relatively small-scale, immediate actions that needed be taken, and clear in general terms about the end-state to be achieved (a net zero city), but the all-important medium term, leading from one to the other, is still an enigma, an unmapped and unknown territory. A phronetic approach will help both practitioners and researchers to navigate this difficult medium term through a process of ongoing learning. The path to delivering radical emission reductions may remain shrouded in fog, but the process of learning how to navigate the first steps of the path, enacting new narratives about climate action and its relations with existing responsibilities, priorities and lived experience, will in itself build a capacity for navigating later steps. Moreover, the need for such an approach will only be intensified by the aftermath of the more experientially immediate COVID-19 crisis, when councils will inevitably face budget cuts and pressure to retrench and focus on delivering core statutory services, and will face an external environment in which the following applies:

> *"Your external stakeholders who are basically saying, 'You are out of touch, you have no idea, I have no money in the bank, my business has closed and you're talking to me about carbon."*

A phronetic approach will thus be vital for local government in playing both a leadership and an implementation role in rapid climate action, and for future research to support them in this mission. How exactly to put it into practice in any particular context, however, is itself a matter of the situated phronetic judgement that the approach is trying to cultivate, and so it is all but impossible, by definition, to set out in advance. Hence, while at this stage when this shift in approach is still being presented anew, illustrative examples are hard to come by, our expectation is that attempts by local government stakeholders *themselves* to use the personas sketched out here as a framework for organisational, team or individual action would quickly develop a body of learning that would generate significant momentum, and especially where such lessons are shared across other local government institutions.

As researchers employing a phronetic approach, we feel that it is important to reflect on our own roles, our expectations of the research, and our reactions to the process and outcomes. A significant finding here was that, in conducting the interviews and workshops, we found it more difficult than we had anticipated to encourage participants to step away from their professional roles or 'corporate' positions. Initially, in our conversations with them, participants would stick to the official account of their organisation's performance. When prompted, some did reflect on their own roles, and started to talk through the difference between what they presented in public—the 'official line'—and their private uncertainties. We found that the workshop format, and particularly small group discussions and the creative visualization, helped to uncover these perspectives.

## 5. Conclusions and Implications for Policy

This research did not aim to develop detailed policy recommendations or prescriptions. However, our analysis points to some ways forward for the government at both a local and national level, which would help local decision-makers to implement rapid climate action.

First, it is important for the government, at both a national and local level, to acknowledge the rapid and far-reaching nature of the change that is needed. This allows a more open and honest debate about the 'implementation gap', and the fact that new ways of working will be necessary. An acknowledgement of the scale and nature of the issue frees up all parts of an organisation to respond to the challenge, and be upfront about the potential clash with existing procedures, priorities, policies and strategies. In practical terms, such acknowledgement would consist of a clear, unequivocal message from the national government about the need for rapid, co-ordinated climate action, led by all parts of the government.

Second, a common theme was the need for the national government to set a framework for local areas, making clear their responsibilities on climate, and resourcing them to respond, whilst leaving flexibility to allow local areas to develop their own responses. It is increasingly clear that some key levers (e.g., of fiscal or industrial policy) regarding the kind of profound change needed to address climate emergency reside squarely and solely in the hands of the nation-state. A wholesale agenda of the devolution of responsibility to local government could well thus do little to advance deep decarbonisation. Yet building the capacities of local government, and especially through and for ongoing phronetic learning, could have a doubly positive effect, enabling a new productive and cooperative division of responsibility for climate action between local and national government.

On the one hand, a more capable and empowered local government could increase and sustain pressure on the national government to use those essential but currently neglected levers that are uniquely at its disposal. On the other, though, the national government could be more willing to devolve other relevant powers to local government since this turns out to be a more effective way to deliver on national government priorities and manifesto promises, and crucially, without diminishing, but possibly enhancing, the power of the national (centralised) government. In short, a deeper engagement with a phronetic approach could empower local government to demand of the national government not that it hand over the reins, but, almost to the contrary, that it summons the political will to actually use the unique powers of the nation-state on this agenda *in parallel with* a greater

devolution of other responsibilities to the local level. This could, for example, take the form of a 'devolution deal' for climate, to provide a clear specification of the division of responsibilities between the national and local government.

Third, cities and other local areas should be prepared for the overall aim of responding to the climate emergency to conflict with existing procedures, referred to as the 'devil in the detail'. Local areas could create a mechanism that would allow local officers or politicians to flag such conflicts, and work through their implications and potential solutions, rather than—as is often the case currently—trying to work around them. In practice, this would mean local leaders making changes both to administrative procedures and to political priorities, to emphasise that climate action is a priority, and encourage teams to address conflicts rather than working around them.

Lastly, our study has highlighted the vital role played by local politicians and officials, using their own experience and understandings to develop and advocate ways forward. The participants in this project found that their involvement, and the opportunity that enabled them to reflect on the challenges and dilemmas they faced through the interviews and the city workshops, was helpful. This sort of support could be provided more widely, for example through training programmes, separate from existing systems of management or strategy development. Following the presentation of our findings to the local government officials, we have had enquiries about how it could be used in this way. This could help to develop working cultures which allow for a full and frank discussion about how best to respond to the climate emergency.

**Author Contributions:** Conceptualisation, methodology, writing, editing and project administration: A.Y., D.T. and R.W. All authors have read and agreed to the published version of the manuscript.

**Funding:** This research was funded by the UK's Economic and Social Research Council (ESRC) through the Place-Based Climate Action Network (P-CAN), grant number ES/S008381/1.

**Institutional Review Board Statement:** The study was conducted according to the guidelines of the Declaration of Helsinki, and approved by the Lancaster University Faculty of Science and Technology Ethics Committee (protocol code FST19086, approved 20 February 2020).

**Informed Consent Statement:** Informed consent was obtained from all subjects involved in the study. Written informed consent has been obtained from the participants to publish this paper.

**Data Availability Statement:** The data in this study consists of transcripts from interviews and workshops. These cannot be made available because of the ethics code agreed, and the need to assure anonymity for participants.

**Acknowledgments:** The authors would like to thank all those who took part in interviews and workshops, as well as Candice Howarth and Kate Lock from PCAN.

**Conflicts of Interest:** The authors declare no conflict of interest. The funders had no role in the design of the study; in the collection, analyses, or interpretation of data; in the writing of the manuscript, or in the decision to publish the results.

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
