# Peer review of "Implementing Rapid Climate Action: Learning from the ‘Practical Wisdom’ of Local Decision-Makers"

_sustainability, doi:10.3390/su13105687_

Round 1

Reviewer 1 Report

Dear Authors,

Generally, the paper is very interesting. I think the idea of drawing conclusions from interviews with local decision-makers is accurate and interesting. Today, humanity has amazing capabilities to counteract climate change. The question is whether we are prepared for it, and whether we want to. Still, many important decisions remain with the government. This article seems to fill a void in this regard. Thus, it provides a lot of interesting information.

However, I must admit that the paper is a bit unusual for this magazine. Despite this, I think that it fits well with the idea of sustainable development and will be a nice supplement to the knowledge about counteracting climate change.

Below are some notes:

Line 208: The country in which the city is located should be added.

In turn, when naming universities, the city should be added next to each of them.

Line 221: In my opinion, the work should list selected questions that have been asked to stakeholders (at least in general terms). This would allow a better understanding of the analyzes carried out.

Author Response

Generally, the paper is very interesting. I think the idea of drawing conclusions from interviews with local decision-makers is accurate and interesting. Today, humanity has amazing capabilities to counteract climate change. The question is whether we are prepared for it, and whether we want to. Still, many important decisions remain with the government. This article seems to fill a void in this regard. Thus, it provides a lot of interesting information.

Reply: We thank the referee for their supportive comments regarding the article

However, I must admit that the paper is a bit unusual for this magazine. Despite this, I think that it fits well with the idea of sustainable development and will be a nice supplement to the knowledge about counteracting climate change.

Reply: We thank the referee for their supportive comments regarding the article

Below are some notes:

Line 208: The country in which the city is located should be added.

Reply: added, as requested

In turn, when naming universities, the city should be added next to each of them.

Reply: added, as requested (e.g. in author addresses)

Line 221: In my opinion, the work should list selected questions that have been asked to stakeholders (at least in general terms). This would allow a better understanding of the analyzes carried out.

Reply: at line 279, we have added further clarification, in general terms, of the questions put to the stakeholders involved.

Reviewer 2 Report

I have enjoyed reading this manuscript. The authors’ have conducted an in-depth investigation of the gap between climate goals and policy implementation and the ‘phronetic knowledge’ of the three local governments in this study. The results are clearly explained and provide valuable feedback from the climate policy experiences of local officials and representatives. The authors’ decision to utilize qualitative methods to examine this topic provides insights both for readers and the research participants. This study helps shed light on an important challenge that remains a barrier to successfully implementing climate policies and adds knowledge to the academic literature and helpful advice to decision-makers.

Suggestions for revisions:

- Results sections 3.1 and 3.2: in addition to the descriptive text of results, consider a table to display the priorities and personas in a brief snapshot that the reader can quickly skim; visuals and graphics can help convey dense information more effectively for many readers

pg. 5, line 217: correct misspelling of ‘university’

pg. 6, line 288: “or whether local political configurations of made progressive 288 action easier or harder.” this statement is confusing, needs rewording perhaps

pg. 11, line 504: “changed” should be “change”

Author Response

I have enjoyed reading this manuscript. The authors’ have conducted an in-depth investigation of the gap between climate goals and policy implementation and the ‘phronetic knowledge’ of the three local governments in this study. The results are clearly explained and provide valuable feedback from the climate policy experiences of local officials and representatives. The authors’ decision to utilize qualitative methods to examine this topic provides insights both for readers and the research participants. This study helps shed light on an important challenge that remains a barrier to successfully implementing climate policies and adds knowledge to the academic literature and helpful advice to decision-makers.

Reply: We thank the referee for their supportive comments regarding the article

Suggestions for revisions:

- Results sections 3.1 and 3.2: in addition to the descriptive text of results, consider a table to display the priorities and personas in a brief snapshot that the reader can quickly skim; visuals and graphics can help convey dense information more effectively for many readers

Reply: We thank the referee for this suggestion, and have added the tables – now Tables 2 and 3

pg. 5, line 217: correct misspelling of ‘university’

Reply: Corrected.

pg. 6, line 288: “or whether local political configurations of made progressive 288 action easier or harder.” this statement is confusing, needs rewording perhaps

Reply: Corrected, to read “or whether local political configurations have made progressive…”

pg. 11, line 504: “changed” should be “change”

Reply: Corrected.

Reviewer 3 Report

Theoretical background is part of the methodology, which is not usually the standard in scientific articles.
The literature is relatively poorly processed, it would be appropriate to include in the literature the climate strategies of selected countries.
Although the results include the results of qualitative research, it is only an evaluation of issues without a broader context.
In the research part, the strategy that the individual analyzed cities have adopted should appear and compare its fulfillment through quantitative indicators.
The whole article present as a sociological probe, without exact link to the adopted strategies in individual towns.
The article is described very generally, it is not stated whether the given cities have some climate strategies and whether they fulfill them in some way. Rather, it is only a subjective evaluation of a relatively small sample of people.
It is necessary to rework the article mainly from the theoretical point of view but also from the practical point of view. There is a lack information of exact scientific resources about strategic climate and climate change management.
The article constantly states the complicated situation in the field of climate change at the local level, but nowhere are the competencies that cities should provide in this area and the tools used by the local government to address the situation are not described. 

Author Response

Theoretical background is part of the methodology, which is not usually the standard in scientific articles.

Reply: We thank the referee for this comment, but note that, this being an article primarily in the qualitative social sciences and with its focus on exploring not what is/must be done but how, the theoretical background explained in this section is, in fact, directly related to questions of methodology.  Hence, for instance, the focus of this discussion around the usual choice between “one of two broad methodologies” and the article’s advocacy of a third.  This is also the most relevant section in which to discuss these key issues for the article.  Hence we have left this sub-section where it is.

The literature is relatively poorly processed, it would be appropriate to include in the literature the climate strategies of selected countries. Although the results include the results of qualitative research, it is only an evaluation of issues without a broader context.

Reply: We thank the referee for this suggestion.  We would ideally like to be able to refer to climate strategies of selected countries, but as the focus of this paper (and, indeed, special issue as a whole) is on local government, this discussion would have to be either long and involved – distracting from the point of this article and taking up a lot of word count – or not sufficiently detailed to get below national-level policies to the level of government here in question.  We have added, however, a whole new discussion (in section 2.2) which contextualizes the situation of local government regarding climate action in the UK, which will enable comparison by readers with situations regarding local government in other countries.

In the research part, the strategy that the individual analyzed cities have adopted should appear and compare its fulfillment through quantitative indicators. The whole article present as a sociological probe, without exact link to the adopted strategies in individual towns.

Reply: We thank the referee for these helpful suggestions.  We have added a new table (Table 1) which summarizes the climate action plans in each city, as requested. We also note there that “it is only very recently that this has become a significant priority for them, and in most cases plans and strategies are only now emerging, so their impacts cannot yet be quantitatively assessed”; and that “The phronetic approach is not intended to assess the appropriateness of these strategies or targets, nor to measure progress towards implementing or achieving them, but rather to explore how, in practice, key actors within these institutions engage with them and integrate them within everyday working practices that are also subject to a wide range of other pressures, priorities and responsibilities.”

The article is described very generally, it is not stated whether the given cities have some climate strategies and whether they fulfill them in some way. Rather, it is only a subjective evaluation of a relatively small sample of people.

Reply: We thank the referee for this helpful comment. In response, we have added discussion, including in a table, of precisely such strategies in the cities investigated. But we also note that the focus of this research is emphasized at many points as being a matter of ‘how’ not ‘what’ regarding climate action at the local level of government, and that this involved concerted work using qualitative research approaches with small numbers of relevant individuals.  It is thus incorrect to call the research ‘subjective evaluation’ and where the small sample is a supposed weakness of the research since the whole point of the shift in research approach and collective learning is precisely to engender (‘subjective’) insights from the stakeholders themselves, which this article then summarizes.

It is necessary to rework the article mainly from the theoretical point of view but also from the practical point of view. There is a lack information of exact scientific resources about strategic climate and climate change management.

Reply: We thank the referee for this comment, but again note that the focus of this article is explicitly NOT on the ‘what’ (i.e ‘information of exact scientific resources… climate change management’) but the ‘how’ regarding effective climate action, i.e. with a focus on how those actually tasked with these complex challenges can best work together to work through them, not to ‘manage’ or ‘solve’ them. We also refer the referee to the argument in the ‘theoretical orientation’ section that a key problem for climate action, and governance thereof, is that there is, in fact, no definitive and singular ‘fact of the matter’ for science to first ‘get right’ before handing over to practical decisions or policy.  We therefore cannot rework the article as requested without defeating precisely a key argument of the paper as a whole.

The article constantly states the complicated situation in the field of climate change at the local level, but nowhere are the competencies that cities should provide in this area and the tools used by the local government to address the situation are not described. 

Reply: We thank the referee for this comment, but again note that the focus of this article is explicitly NOT on ‘what’ (i.e. questions of competencies and tools) but ‘how’ regarding effective climate action.

Reviewer 4 Report

This paper is very interesting and topical on a major issue. It is well written and clearly presented. My areas for improvement are twofold and are as follows:

Introduction

The introduction should not contain so many results and conclusions that can be presented later in the paper, after a more detailed presentation of the method, actors, etc. In this introduction, it would be more useful to present the organizational context and governance of these cities and how the interviewees position themselves in this organization (before entering the subject of understanding the context of decision-making, the purpose of the study). The theoretical orientation is well explained in the methodology but not the "organizational ground" in which the actors interviewed operate, to better understand for example the levers and brakes that may be inherent in their governance and also influence decisions. And similarly, what types and degrees of action should be expected at this level compared to another decision scale. In other words, just to help us understand and become aware of the system in which they operate. This is notably true since it is stated in the introduction, L96-97, “through a conflicting and sometimes contradictory governance landscape.” (And further on, which is also reflected in the results with national-local interactions).

Operationalization of solutions

Beyond the awareness of this typology and its usefulness for better action, how can this typology be effectively considered by the actors in the exercise of their functions? What recommendations do you have for fitting this into their decision-making and action implementation patterns?

L519-523: “For example, at an institutional level it could be used to map and develop competencies, aims and ways of working within and across departments, and to shape corporate and communication strategies on climate action; at a team or individual level it could be used to scope out the most appropriate personas to engage at different stages of a project and with different stakeholders, to identify gaps in capacity or adjust operational tactics.” : do you have any recommendations on how to do this?

L524-532: the authors propose the application of a matrix but do not give any indication of how this can be operationalized. Some more concrete examples might be welcome to go further and make a greater contribution to these issues. Especially since, as the authors point out “practical implications are not yet grasped and, perhaps more significantly, the path to 554 understanding and engaging with these implications is very far from clear.” (L554-555).

How to achieve this: “the process of learning how to navigate the first steps of the path, enacting new narratives about climate action and its relations with existing responsibilities, priorities and lived experience, will in itself build capacity for navigating later steps” (L594-597) (or more generally how to implement on the ground the - very interesting - approach you advocate?)

in the recommendations proposed in the conclusion, I did not perceive what would allow the phronetic approach to be implemented in these instances, or the application of the matrix for the categories of actors (“In short, a deeper engagement with a phronetic approach could empower local government to demand of national government not that it hand over the reins, but, almost to the contrary, that it summons the political will actually to use the unique powers of the nation-state on this agenda in parallel with a greater devolution of other responsibilities to the local level”. (L643-647))

This does not detract from the relevance of the other recommendations….

Beyond the parallel with the equalities agenda (L556) has the issue of coordinated, multi-issue measures (such as the SDGs) been raised?

L42: (p. 3)?

Author Response

This paper is very interesting and topical on a major issue. It is well written and clearly presented.

Reply: We thank the referee for their supportive comments regarding the article

My areas for improvement are twofold and are as follows:

Introduction

The introduction should not contain so many results and conclusions that can be presented later in the paper, after a more detailed presentation of the method, actors, etc. In this introduction, it would be more useful to present the organizational context and governance of these cities and how the interviewees position themselves in this organization (before entering the subject of understanding the context of decision-making, the purpose of the study). The theoretical orientation is well explained in the methodology but not the "organizational ground" in which the actors interviewed operate, to better understand for example the levers and brakes that may be inherent in their governance and also influence decisions. And similarly, what types and degrees of action should be expected at this level compared to another decision scale. In other words, just to help us understand and become aware of the system in which they operate. This is notably true since it is stated in the introduction, L96-97, “through a conflicting and sometimes contradictory governance landscape.” (And further on, which is also reflected in the results with national-local interactions).

Reply: We thank the referee for these helpful comments.  In response, we have added a new discussion in section 2.2 setting out the broader context of local government in the UK and its specific responsibilities vis-à-vis national government for climate action, as well as a table (Table 1) summarizing the specific climate governance plans in each of the cities studied.  This new text should help Sustainability’s international readership to contextualize the situation under discussion and compare it for themselves with local government elsewhere in the world. We have also stressed the role of locally-agreed and/or local government-originating climate strategies throughout the text where this emphasis will likely be helpful to readers.

Operationalization of solutions

Beyond the awareness of this typology and its usefulness for better action, how can this typology be effectively considered by the actors in the exercise of their functions? What recommendations do you have for fitting this into their decision-making and action implementation patterns?

Reply: We thank the referee for all the comments and queries about operationalization of the different approach presented. In response, we have added several ideas and suggestions.  Regarding recommendation to fit this typology into decision-making, in the final paragraph we note the possibility of training in use of the typology, and the fact that we have had invitations to offer such training by local governments attending the project’s closing webinar.

L519-523: “For example, at an institutional level it could be used to map and develop competencies, aims and ways of working within and across departments, and to shape corporate and communication strategies on climate action; at a team or individual level it could be used to scope out the most appropriate personas to engage at different stages of a project and with different stakeholders, to identify gaps in capacity or adjust operational tactics.” : do you have any recommendations on how to do this?

Reply: In this case, we find it hard to be more specific about how exactly this suggestion could be done in practice, since more detail would inevitably require more concrete examples to work with – which we do not have to offer at this stage in the research, unfortunately.

L524-532: the authors propose the application of a matrix but do not give any indication of how this can be operationalized. Some more concrete examples might be welcome to go further and make a greater contribution to these issues. Especially since, as the authors point out “practical implications are not yet grasped and, perhaps more significantly, the path to 554 understanding and engaging with these implications is very far from clear.” (L554-555).

Reply: In this case, we would suggest that the operationalization of the table is not the key point being made at this point, in this paragraph.  Rather, the purpose of the matrix is simply to offer a more easily-remembered and understood formulation of the four personas that have been inductively identified.

How to achieve this: “the process of learning how to navigate the first steps of the path, enacting new narratives about climate action and its relations with existing responsibilities, priorities and lived experience, will in itself build capacity for navigating later steps” (L594-597) (or more generally how to implement on the ground the - very interesting - approach you advocate?)

Reply: We have responded to this helpful suggestion in lines 670 et seq. “How exactly to put it into practice in any particular context, however, is itself a matter of the situated phronetic judgement it is trying to cultivate, and so all-but-impossible, by definition, to set out in advance. Hence, while at this stage when this shift in approach is still being presented anew, illustrative examples are hard to come by, our expectation is that attempts by local government stakeholders themselves to use the personas would quickly develop a body of learning that would generate significant momentum; and especially where such lessons are shared across other local government institutions.”

in the recommendations proposed in the conclusion, I did not perceive what would allow the phronetic approach to be implemented in these instances, or the application of the matrix for the categories of actors (“In short, a deeper engagement with a phronetic approach could empower local government to demand of national government not that it hand over the reins, but, almost to the contrary, that it summons the political will actually to use the unique powers of the nation-state on this agenda in parallel with a greater devolution of other responsibilities to the local level”. (L643-647))

Reply: We have revised and added wording on this point (now at lines 718 et seq.) aiming to clarify and illustrate these points further.

This does not detract from the relevance of the other recommendations….

Beyond the parallel with the equalities agenda (L556) has the issue of coordinated, multi-issue measures (such as the SDGs) been raised?

Reply: This analogy was not raised in our research, but we thank the referee for the suggestion and have noted it in the text – see l.628-630

L42: (p. 3)?

Reply: Corrected to reference p.755 of reference [8]

Round 2

Reviewer 3 Report

The article in its current form, after additions and modifications by the authors, is well revised.